# Evidence for the Contribution of Gut Microbiota to Age-Related Anabolic Resistance

**DOI:** 10.3390/nu13020706

**Published:** 2021-02-23

**Authors:** Matthew D. Watson, Brett L. Cross, Gregory J. Grosicki

**Affiliations:** Biodynamics and Human Performance Center, Georgia Southern University (Armstrong Campus), Savannah, GA 31419, USA; mw26350@georgiasouthern.edu (M.D.W.); bc09874@georgiasouthern.edu (B.L.C.)

**Keywords:** sarcopenia, skeletal muscle, protein, aging, muscle protein synthesis

## Abstract

Globally, people 65 years of age and older are the fastest growing segment of the population. Physiological manifestations of the aging process include undesirable changes in body composition, declines in cardiorespiratory fitness, and reductions in skeletal muscle size and function (i.e., sarcopenia) that are independently associated with mortality. Decrements in muscle protein synthetic responses to anabolic stimuli (i.e., anabolic resistance), such as protein feeding or physical activity, are highly characteristic of the aging skeletal muscle phenotype and play a fundamental role in the development of sarcopenia. A more definitive understanding of the mechanisms underlying this age-associated reduction in anabolic responsiveness will help to guide promyogenic and function promoting therapies. Recent studies have provided evidence in support of a bidirectional gut-muscle axis with implications for aging muscle health. This review will examine how age-related changes in gut microbiota composition may impact anabolic response to protein feeding through adverse changes in protein digestion and amino acid absorption, circulating amino acid availability, anabolic hormone production and responsiveness, and intramuscular anabolic signaling. We conclude by reviewing literature describing lifestyle habits suspected to contribute to age-related changes in the microbiome with the goal of identifying evidence-informed strategies to preserve microbial homeostasis, anabolic sensitivity, and skeletal muscle with advancing age.

## 1. Introduction

By the year 2030, all Baby Boomers will be at least 65 years of age, and this portion of the population will comprise more than 20% of individuals residing in the United States [1]. The aging process is associated with pervasive physiological declines that are exemplified by reductions in size and function of skeletal muscle (i.e., sarcopenia) [2]. Given the association between sarcopenia with adverse health outcomes (e.g., falls, fractures, and mobility limitations) and mortality [3], a more definitive understanding of the biological mechanisms underlying sarcopenia is warranted.

The regulation of skeletal muscle mass is dictated by temporal fluctuations in muscle protein synthesis (MPS) and muscle protein breakdown (MPB) [4]. Though the effect of aging on whole body protein turnover was initially a subject of great contention [5], it is now generally accepted that healthy aging is not accompanied by accelerated MPB [6,7]. Furthermore, comparable rates of basal MPS [8] and turnover [9] have been observed in healthy older and younger adults. However, in older animals and humans alike, the sensitivity of MPS to anabolic stimuli, such as protein feeding, is substantially diminished when compared with that in the young [8,10,11]. This blunted anabolic responsiveness, termed anabolic resistance, is highly characteristic of aging skeletal muscle [12], and much effort is being devoted to delineating the etiology of this phenomenon. Greater insight into this area may help to optimize the rehabilitative role of protein intake for the maintenance and/or recovery of skeletal muscle tissue in older adults [6,13].

The gut microbiota refers to the collective of bacteria, archaea, viruses, and eukaryotic microbes that reside in the gastrointestinal tract [14]. Though best known for its role in nutrient uptake, the gut microbiome is also intricately connected to a diverse array of physiological systems; influencing metabolic function, protecting against pathogens, and modulating immune response [15]. Recent studies by several independent research groups have provided evidence for a bidirectional gut-muscle axis with profound implications for aging skeletal muscle and sarcopenia [16,17,18,19,20,21]. As studies supporting a role for the gut microbiome in regulating muscle mass and function continue to accumulate [15,17,20,22,23,24,25,26], whether baseline microbial signatures may influence anabolic potential is deserving of deeper inquiry.

The purpose of this review will be to provide evidence in support of the hypothesis that age-associated changes in gut microbiota composition contribute to anabolic resistance following protein feeding in older adults that underlie sarcopenia. We will begin by outlining how changes in gut microbiota that are hallmark of aging may impact protein digestion and amino acid absorption, reduce circulating amino acid availability, contribute to anabolic hormone deficiencies or impair responsiveness, and play a role in intramuscular signaling deficits - all of which may underlie age-related anabolic resistance. Along the way, the gut microbiota of long-lived individuals (e.g., centenarians) will be considered as this population provides researchers with an insightful model of healthy aging. With these mechanisms and considerations in mind, we will evaluate lifestyle habits suspected to contribute to age-related changes in the microbiome with the goal of identifying evidence-informed strategies to preserve microbial homeostasis and potentially improve anabolic sensitivity in older adults. Finally, we will conclude by identifying important gaps in knowledge that may serve as fertile ground for future research in this area.

## 2. The Aging Gut Microbiome & Anabolic Sensitivity

### 2.1. Overview of the Aging Microbiome, Anabolic Resistance, & Sarcopenia

Although human gut microbiome composition seems relatively stable throughout early life, recent studies have highlighted distinct shifts in microbiota in later years (≥65 yrs) [27,28]. Reductions in microbial diversity and short chain fatty acid (SCFA) producing bacteria, concomitant with pathobiont overgrowth (e.g., streptococci, staphylococci, enterobacteria, and enterococci), as well as rearrangements within the Firmicutes and Bacteroidetes phyla, are highly characteristic of the aging gut microbiome, and greater inter-individual variability in microbial composition is commonly observed among older individuals [29,30]. In this section we will examine the ways in which age-related changes in gut microbiota may impact anabolic sensitivity to protein feeding (Table 1) and compare these age-related changes with those observed in healthy, long-lived individuals, an increasingly explored population that may provide novel perspective regarding the maintenance of anabolic sensitivity and muscle mass. 

### 2.2. Protein Digestion & Amino Acid Absorption

Following a protein-rich meal, effective digestion and absorption of dietary proteins is an essential first step in facilitating a muscle protein synthetic response [72]. Protein digestion begins in the acidic environment of the stomach, where low pH-induced protein denaturation yields more accessible polypeptides for subsequent protein processing by activated pepsin. Once in the small intestine, proteases and peptidases produced and secreted by the pancreas (e.g., trypsin, chymotrypsin, carboxypeptidase) and the intestinal epithelium (e.g., aminopeptidase N) further advance the digestive process. While the majority of dietary protein is absorbed in the small intestine, a small but physiologically relevant quantity (~6–18 g/day) of peptides and amino acids [73] will enter the large intestine for microbial fermentation [74]. 

Given the importance of maximizing circulating amino acid availability to optimize post-prandial MPS [75,76], any defects in the process of protein digestion and absorption may detract from anabolic potential. For example, it has been demonstrated that postprandial amino acid kinetics are affected by age [77]. Moreover, peak plasma appearance of amino acids following a high protein meal is substantially delayed in healthy older vs. younger individuals [77]. Taken together, these findings allude to the possibility that slowed protein digestion and absorption rates in older adults may diminish muscle protein synthetic responses to protein feeding. Identification of the biological mechanisms underlying age-related differences in protein digestion and absorption that appear to coalesce to impair MPS [78] may aid in the development of strategies to maximize anabolic response to protein feeding with aging.

The prominent role of gut microbiota in energy harvest has long been known [79]. In regards to the metabolism of dietary protein, gut microbes are involved in digestive, absorptive, and metabolic processing of amino acids within the gastrointestinal tract [80]. Pioneering work in animal models first highlighted an influence of intestinal microbiota on proteolytic enzyme activity in the small intestine. Conventionalization of germfree (GF) piglets with feces derived from a clinically healthy sow was demonstrated to reduce the activity of aminopeptidase N [31], a brush border enzyme involved in protein hydrolysis. Moreover, monoassociation of GF pigs with commensal *Escherichia coli* strains is demonstrated to induce maturational changes of the intestinal brush border characterized by increases in enzymatic activity (i.e., sucrase, glucoamylase, and aminopeptidase N) [32]. These findings emphasize the significance of gut microbiota in shaping metabolic enzymatic activity in the small intestine (Table 1). Interestingly and somewhat paradoxically, activity of aminopeptidase and other protein digestive enzymes appear to be elevated in senescent rats [81], which may serve to offset age-related declines in gastric pepsin secretion [82]. Whether the microbiome plays a role in this apparent compensatory response, and if a similar phenomenon is observed in aging humans remains to be determined.

A more direct role for gut microbes in protein digestion is observed through the proteolytic fermentation of undigested peptides in the large intestine. Due to the variety of toxic substances this may produce (e.g., hydrogen sulfide, ammonia, *p*-cresyl and indoxyl sulfate), this process is generally considered to be harmful to the host [33]. Indoxyl sulfate is shown to potentiate skeletal muscle atrophy via induction of reactive oxygen species (ROS), inflammatory cytokines, and myoatrophic gene expression [23]. The greater abundance of indoxyl sulfate and other toxic byproducts (*p*-cresyl) observed in older individuals [34] may be attributed to an increase in protein fermenting bacteria such as *Clostridium perfringens, Desulfovibrio, Peptostreptococcus, Acadaminococcus, Veillonella, Propionibacterium, Bacillus, Bacteroides,* and/or *Staphylococcus* [35]. More work is needed to pinpoint the precise microbial dynamics contributing to this age-related increase in toxic microbial metabolites as a means of providing therapeutic targets for future interventional studies.

Contrariwise, SCFA’s and biogenic amines, which are also end products of protein fermentation in the large intestine, serve many important physiologic functions and are thought to provide ~10% of daily energy requirements [83]. Colon-derived SCFA’s also appear to play a role in protein metabolism and anabolic responsiveness [84]. For instance, chronic supplementation with the SCFA butyrate has been reported to prevent hindlimb muscle loss in aging mice [85], and a butyrate-containing SCFA cocktail was recently observed to increase muscle mass in mice lacking a microbiome [25]. Of the mechanisms through which butyrate is purported to influence protein metabolism, the anti-inflammatory benefits of histone deacetylase (HDAC) inhibition is among the most pervasively discussed [85]. Microbial-derived SCFA’s are also demonstrated to promote epithelial barrier function, thereby reducing intestinal permeability and protecting against inflammation [86]. In this context, declines in butyrogenic microflora (e.g., *Roseburia, Clostridia,* and *Eubacteria*) [36] that are characteristic of aging [37] may be viewed as a key contributor to age-related anabolic resistance. Intriguingly, several reports examining the microbiome of healthy, long-lived animal and human models (≥90 years) have observed microbial signatures containing elevated levels of SCFA producers [87,88,89]. Interpreted in unison, these findings highlight that age-related changes to protein digestion and absorption in the small intestine, via microbial interactions with proteolytic enzymes, may contribute to anabolic responsiveness. Nevertheless, the microbiome may have its greatest impact on protein metabolism within the large intestine through the production of various protein metabolites with implications for down-stream protein synthetic processes, which will be discussed in greater detail in future sections.

### 2.3. Circulating Amino Acid Availability

Postprandial circulating amino acid kinetics profoundly influence whole body protein anabolism [90]. To overcome age-related deficits in protein digestion and absorption and to maintain circulating amino acid availability, older individuals are advised to ingest a daily protein quantity that is in excess of the Recommended Dietary Allowance [91]. While this elevated protein intake may help to attenuate declines in circulating amino acid availability, it is also important to consider the contribution of microbial-derived amino acids to the circulating amino acid pool [42,92]. Relevantly, microorganisms such as Bifidobacteria and Clostridia, which decline during late adulthood [93], yet are observed to increase in healthy 90+ year old individuals [88,89,94,95,96,97], are demonstrated to produce physiologically relevant amino acids from nonspecific nitrogen sources [98,99]. Preliminary evidence for the *de novo* biosynthesis of amino acids (i.e., lysine) by the gut microflora *in vivo* was initially provided by comparative experiments using the ^15^N labeling paradigm in GF and conventionalized rats [38]. This finding has since been confirmed in human studies suggesting that as much as 20% of circulating plasma lysine is derived from intestinal microbial sources [39]. Relevantly, lysine is an essential ketogenic amino acid that plays a fundamental role in muscle protein turnover [100], and high lysine levels have even been linked to human longevity [101]. Interestingly, human aging is associated with a reduction in bacteria such as *Prevotella* [40] that are involved in lysine biosynthesis [41], and this decline is predictive of physical frailty [102] as well as loss of independence [103]. Perhaps more important than the microbial contribution to the total body supply of lysine, intestinal microbes are estimated to synthesize between 19 to 22% of leucine [42], an essential amino acid that is speculated to play a unique role in the initiation of anabolic intracellular signaling pathways [104]. Longitudinal studies are needed to confirm cross-sectional observations implicating bacteria such as *Prevotella, Allistiples, and Barnesiella* in the biosynthesis of leucine and other anabolic amino acids [43].

Detracting from circulating amino acid availability, splanchnic extraction of dietary amino acids increases with age [44,105,106], and this phenomenon appears to be exacerbated by obesity [106]. Greater first-pass splanchnic amino acid uptake in older individuals may be explained by increased leucine oxidation in the gut and/or liver [45], as is observed in healthy older men who fail to suppress leucine oxidation in response to experimental hyperglycemia and/or following exercise training [46]. This reduced metabolic flexibility, characterized by a greater reliance on amino acid metabolism, may be influenced by microflora. For example, antimicrobial treatment is demonstrated to suppress leucine oxidation (~15–20%), conceivably through the elimination of subclinical infections [42,47] brought about by maladaptive changes in the gut microbiome [107]. With this in mind, it is intriguing to speculate as to the contribution of age-related microbial dysbiosis to heightened rates of leucine oxidation and thus greater splanchnic sequestration of amino acids (Table 1). To assuage age-related inefficiencies in amino acid availability and utilization, therapies to promote myogenic microbial-derived amino acids and to combat maladaptive microbial alterations that are associated with aberrant changes in protein metabolism are of interest.

### 2.4. Anabolic Hormone Responses

The endocrine system plays a fundamental role in the regulation of muscle mass. Insulin, growth hormone (GH), and insulin-like growth factor 1 (IGF-1), are responsive to exogenous amino acid intake and are shown to influence muscle growth and development throughout life [108]. However, the relevance of transient, systemic elevations of these hormones following protein feeding and/or mechanical stimulation for the promotion of muscle growth is challenged by studies demonstrating muscle hypertrophy [109], intramuscular signaling [110], and MPS [111] independent of endocrine responses. These findings have led to the consensus that endocrine responses following anabolic stimuli appear to play a permissive, rather than stimulatory, role in mediating MPS [108,112]. Nonetheless, associations have been made between age-related changes in body composition with insulin resistance [113,114], and declining GH and IGF-1 levels [115,116,117,118,119]. Collectively, these observations allude to the relevance of anabolic hormonal deficiencies in mediating aging muscle loss, and provide incentive for research examining the etiology of these changes in endocrine regulation.

Conventionally, it is understood that insulin acts upon skeletal muscle to facilitate glucose uptake [120], and plays a permissive role in MPS by upregulating downstream anabolic signaling events [121]. Less commonly discussed is insulin’s anabolic actions through an upregulation of skeletal muscle blood flow, an important determinant of MPS deserving of greater examination due to its prospective relationship with the aging gut microbiome. Though it is currently unclear whether insulin-mediated MPS is independent from [48,122,123], or coincides with [124,125,126], skeletal muscle glucose tolerance, past studies have highlighted how insulin may mediate vasodilation through its influence on endothelial function [114,127,128]. Therefore, it is intriguing to speculate how age-related impairments in insulin-mediated tissue perfusion may detract from anabolic responsiveness.

Recent work by Fujita et al. [129] demonstrated that experimental hyperinsulinemia promotes MPS in a manner that is best predicted by insulin-related changes in skeletal muscle blood flow. Corroborating this observation, Nygren and Nair [130] demonstrated that concurrent insulin and amino acid infusion augments MPS in healthy young adults. The mechanism underlying this insulin-mediated blood flow augmentation was outlined in landmark studies by Steinberg et al. [131] and Scherrer et al. [123], where insulin-mediated activation of endothelial nitric oxide synthase (eNOS) was observed. However, experimental hyperinsulinemia has been demonstrated to increase vastus lateralis blood flow in younger but not older (≥65 years), healthy individuals [122]. 

Aging research suggests that reductions in nitric oxide (NO) biosynthesis and bioavailability, rather than decreased endothelial NO sensitivity, may instigate the attenuated insulin-mediated skeletal muscle perfusion that is hallmark in aging muscle [48,50,51,128]. There is growing support that gut microbes may impair vascular function and contribute to this phenomenon [48,49,50,132]. Specifically, increases in inflammation, oxidative stress and ROS owing to maladaptive changes in the gut microbiome all may limit eNOS release in the presence of insulin [48,49,50,51,52,132]. Meanwhile, antibiotic treatment in aging mice has been demonstrated to suppress gut dysbiosis through the attenuation of pathogenic microbes (e.g., Proteobacteria, Verrucomicrobia, and Desulfovibrio) and reduce inflammation, facilitating improvements in vascular function [48,49]. Though a comprehensive understanding of the ways in which the microbiome seems to impair endothelial insulin sensitivity with age is lacking, circulating microbial metabolites such as trimethylamine-*N*-oxide (TMAO) and uremic toxins (e.g., indoxyl sulfate) are postulated to play a role.

Production of TMAO begins with microbial conversion of substrate, most notably choline, to trimethylamine (TMA), which is readily absorbed and quickly converted to TMAO in the liver [51]. In older individuals and clinical populations, elevated levels of TMAO are associated with inflammation and ROS that inhibit eNOS production [49,50,52]. Pertinently, age-related increases in plasma TMAO levels are associated with decreased NO bioavailability and vascular dysfunction [50,52]. For example, in a study by Brunt et al. [50], plasma choline and TMAO levels were inversely related to NO-mediated endothelial-derived dilation [50]. Intriguingly, elevated TMAO levels in old mice are attenuated by antibiotic treatment [49], seemingly through the elimination of known TMAO producers such as select genera from the Firmicutes, Actinobacteria and Proteobacteria phyla [51,52]. In comparison, recent work by Hallberg et al. [133] noted that healthy centenarians exhibit a greater abundance of *Eubacterium limosum*, which have the potential to demethylate TMA, preventing its conversion to circulating TMAO. Mentioned earlier, the uremic toxin indoxyl sulfate may compound TMAO-mediated vascular dysfunction by potentiating ROS production [53]. Synthesis of the above [48,49,50,51,52,53,132,133] highlights how gut dysbiosis and resultant deleterious microbial metabolites, such as TMAO and indoxyl sulfate, may lead to elevated levels of circulating inflammation and ROS, which hinder insulin-mediated skeletal muscle perfusion and impair muscle protein synthetic response to protein feeding with age (Table 1).

It is proposed that GH may not directly influence skeletal muscle anabolism [134,135,136], and instead its effects are mediated through an IGF-1 stimulated anabolic signaling cascade [135,136,137]. Wan et al. [138] noted that piglets fed a low protein diet displayed 50% less serum IGF-1 than those fed a crude protein diet, and also exhibited diminished levels of IGF-1 mRNA expression. Although we were not able to identify any direct comparisons of IGF-1 responses to protein feeding in younger and older humans, Dillon et al. [139] reported that muscle-specific IGF-1 content was increased in healthy older women following 3 months of essential amino acid supplementation. Moreover, several human studies have displayed relationships between habitual protein intake and serum IGF-1 levels [140,141,142]. Collectively, these data provide support for the idea that habitual protein intake influences IGF-1 levels, a finding made relevant by established associations between IGF-1 and both appendicular muscle mass and sarcopenia [143].

The gut microbiome has emerged as a viable mediator of IGF-1 activity. Supporting this notion, Yan et al. [144] demonstrated that conventionalization of GF mice upregulates IGF-1 release, while antibiotic treatment of wild type mice induces IGF-1 downregulation. Moreover, in suckling pigs, faecal microbiota transplantation was associated with an in-crease in GH and IGF-1 levels concurrent with proliferation of Lactobacillus spp. and an increase in the SCFA’s acetate and butyrate [145]. Though the specific microbial species involved in modulating this apparent gut-IGF-1 relationship remain to be determined, *Lactobacillus plantarum (L. plantarum)* species have garnered support [54,55,56,57]. In a nutrient deficient environment, *L. plantarum* rescued *Drosophila* maturation through an apparent upregulation of IGF-1 analogs [54]. Furthermore, monoassociation of GF infant mice in a state of undernutrition with select strains of *L. plantarum* was demonstrated to be both necessary and sufficient to boost postnatal growth via restoration of GH sensitivity and subsequent hepatic IGF-1 release [55]. More recently, it has also been displayed that postbiotic supplementation of *L. plantarum* leads to significantly increased total body weight, butyrate abundance, and hepatic mRNA IGF-1 expression in both heat-stressed broiler chickens [56] and post-weaning lambs [57]. Relevantly, associations between advancing age and reductions in *Lactobacillus* genera are consistently observed [58,59], but unfortunately these works lacked sufficient resolution to confirm a reduction of *L. plantarum* species. Nevertheless, recent interventional studies show probiotic administration of *L. plantarum* to fortify mucosal integrity [60] and improve exercise performance (i.e., grip strength and swim endurance) with age [61]. Moreover, a recent systematic review underscored the greater abundance of *L. plantarum* in healthy long-lived individuals [146], speculatively highlighting the importance of *L. plantarum* preservation throughout aging. In sum, these studies [54,55,56,57,58,59,60,61,146] provide compelling evidence in support of a role for the gut microbiome, and specifically *L. plantarum*, in influencing the somatotropic axis and thus, skeletal muscle responsiveness. Due to its relationships with markers of muscle performance and the knowledge that *Lactobacilli* abundance decreases with age, yet seem to be preserved in heathy centenarians, maintenance of *L. plantarum* may serve as a therapeutic target in the preservation of muscle mass and avoidance of sarcopenia.

The culmination of this knowledge suggests that GH, and specifically IGF-1, elicit anabolic effects in response to protein intake and exhibit a bidirectional relationship with the gut microbiome (Table 1). Dysregulation of the IGF-1 anabolic signaling cascade and resultant sarcopenia may be caused by dysbiosis of the gut microbiome and depletion of IGF-1-related microbes such as lactobacilli and speculatively *L. plantarum*. Preservation of these advantageous microbial strains is an area of anabolic resistance research that is deserving of future investigation in human subjects.

### 2.5. Intramuscular Signaling

The intricate balance between MPS and MPB, which drives skeletal muscle mass maintenance, is ultimately dictated by underlying molecular mechanisms. One of the most widely accepted mediators of protein synthesis is the mammalian target of rapamycin (mTOR) [147]. mTOR stimulates protein synthesis via two actions: 1) The phosphorylation and inactivation of eukaryotic initiation factor 4E-binding protein (4E-BP1), the repressor of mRNA translation, and 2) The phosphorylation and activation of ribosomal S6 kinase (S6K1) [148]. In concert, these molecular events allow protein translation to proceed in an unimpeded fashion. Given the fundamental importance of mTOR signaling in the regulation of MPS, and the demonstrated contribution of down-regulated mTOR activity to aging muscle loss and sarcopenia [149], numerous studies have sought to define the factors underlying mTOR activation. While these investigations have definitively elucidated the role of mechanical stimulation in mTOR regulation, the effects of growth factors and nutrient concentrations on mTOR-mediated protein anabolism warrant more attention.

Growth factors, such as IGF-1, as well as amino acids have been demonstrated to activate mTOR signaling through a complex series of mechanisms [64,150,151,152] outlined in Figure 1. Meanwhile, nutrient deprivation suppresses mTOR activity via phosphorylation and activation of adenosine monophosphate activated protein kinase (AMPK) [63]. Taken together, these findings highlight the importance of nutrient availability and growth factor abundance, both of which may be potently modified by gut microbiota as described above, in arbitrating the molecular regulation of lean tissue mass via mTOR expression (Figure 1).

There are many mechanisms through which gut dysbiosis is posited to contribute to muscle aging and sarcopenia. However, an age-related decrease in epithelial integrity and a corresponding increase in circulating lipopolysaccharides (LPS), owing to a reduction in SCFA producing bacteria, evokes inflammatory signaling and a reduction in insulin sensitivity that may be specifically relevant in the context of the mTOR pathway [21]. Firstly, reduced insulin sensitivity detracts from IGF-1 mediated mTOR stimulation [62]. Secondly, elevated LPS levels may be a key determinant of chronic inflammation, which is associated with reduced adaptation of aging skeletal muscle [65]. Mechanistically, increased expression of pro-inflammatory cytokines are shown to inhibit the mTOR pathway through the activation of AMPK [66,67,68] (Figure 1). Amplifying LPS-mediated inflammatory burden, age-associated ROS over-production may also blunt mTOR signaling [69]. Intriguingly, a mechanism by which enteric microbiota may influence ROS generation has also been proposed [70]. Moreover, communication between dysbiotic microbial communities and mitochondria [71] may exacerbate mitochondrial ROS generation with age. Collectively, these data suggest that adverse changes in the microbiome seen with age contribute to elevated levels of circulating inflammation and ROS that are demonstrated to impair MPS via suppression of skeletal muscle mTOR signaling (Figure 1). However, it is interesting to note that LPS stimulation seems to evoke less of an inflammatory response in long-lived compared to normally age mice [153], alluding to the importance of heightened antioxidant defense mechanisms as a characteristic of highly successful aging.

## 3. Lifestyle Factors Contributing to Age-Related Microbiome Changes

### 3.1. Diet

Dietary habits play an important role in shaping host microbial composition. Proteins, fats, carbohydrates, probiotics, and polyphenols have all been independently demonstrated to induce shifts in the microbiome with meaningful implications for organismal function [154]. Diet-induced modulation of the microbiome can occur in a direct manner, in which nutrients interact with microorganisms to promote or inhibit their growth, or through an indirect influence on immunological and/or metabolic function [155]. In the context of aging skeletal muscle, dietary imbalances and increased intestinal transit time manifest as reductions in microbial diversity and increased intestinal permeability, triggering cytokine-mediated suppression of protein anabolism [156].

Comparison of dietary patterns and microbiome composition in younger and older individuals provides insight into the specific dietary changes contributing to age-related microbial imbalances. As a whole, older adults are more likely to consume an inadequate amount of protein, fruits and vegetables, and total calories when compared to younger individuals [156]. Of these dietary changes, reductions in fiber-containing fruits and vegetables that positively modulate SCFA-producing *Lactobacillus, Prevotella,* and *Bacteroides* genera, which tend to be enriched in healthy long-lived individuals [96,97,146], may be of greatest relevance for anabolic responsiveness (Figure 2). However, considering and correcting for protein-energy malnutrition and other common age-related nutrient deficiencies (e.g., iron, B vitamins, vitamin D) may be necessary to preserve microbial equilibrium [157]. Further, prebiotic and/or probiotic supplementation (e.g., *L. plantarum*) may serve as an additional avenue to improve intestinal homeostasis in older years, although definitive evidence supporting the efficacy of such therapies in human subjects is limited [158,159,160,161].

### 3.2. Exercise

The remarkable plasticity of the human body is exemplified through the rapid and robust adaptation of diverse physiological systems in response to physical activity [162,163,164,165]. Recent research has demonstrated that this exercise-related physiological remodeling is not exclusive to host physiology, but rather extends to commensal organisms within the gastrointestinal tract [166,167,168]. Notably, activity-induced microbial reconfiguration can occur independent of age, though preclinical studies suggest that adaptations may be blunted in older rats [169,170]. To date, the majority of research on this topic has sought to characterize the effects of aerobic exercise on the microbiome [171,172,173,174]. Among these studies, three notable alterations in the gut microbiome have been consistently observed: (1) Increased biodiversity, (2) Greater production of SCFA’s, such as butyrate, and (3) Enhanced gut motility [171,172,173,174,175].

Age-related reductions in gut biodiversity are implicated in heightened levels of circulating inflammation that detract from anabolic responsiveness [103,167,168,176,177]. Meanwhile, exercise-induced increases in microbial diversity are associated with improvements in metabolic homeostasis and reductions in circulating inflammation, though response heterogeneity may be observed [103,166,167,168,178]. Relevantly, Campbell and colleagues [179] demonstrated that this activity-induced increase in gut microbial diversity can occur independent of dietary modification, suggesting that participation in physical activity alone may play a role in therapeutically addressing age-related anabolic resistance. 

Exercise-mediated increases in SCFA producing taxa (e.g., *Clostridiales*, *Roseburia*, *Lachnospiraceae*, and *Erysipelotrichaceae*) may also play a role in addressing age-related anabolic insensitivity [172,180]. These particular microflorae are responsible for producing the SCFA butyrate, which is the primary nutrient for cells of the intestinal epithelium [180]. Butyrate, along with other primary fatty acids, serves to moderate energy balance, nutrient availability, gut motility, and inflammation in the body [166,181,182,183], all of which play a critical role in promoting anabolic response [36,37,91]. Taken together, these studies speak to the plausibility of physical activity as a viable therapy for the age-related gut dysbiosis that is speculated to play a role in the development of anabolic resistance and sarcopenia (Figure 2).

### 3.3. Sleep

Approximately half of older adults report difficulty initiating or maintaining sleep [184]. These complaints are supported by literature demonstrating reductions in total sleep time, more frequent night time awakenings, and decreased slow wave (deep sleep) in older individuals [185]. Recent cross-sectional [186,187,188] and longitudinal [189,190,191] studies have highlighted an intriguing relationship between human gut microbiome composition and sleep physiology. Middle-age patients with obstructive sleep apnea exhibit gut microbial dysbiosis that is characterized by functional deficiencies hallmark of aging – reduced butyrate production and elevated inflammatory biomarkers (i.e., interleukin-6) [191]. In older adults, poorer sleep quality is associated with microbial alterations that are seen with metabolic dysfunction and obesity [188]. Corroborating this observation, short term sleep deprivation is demonstrated to induce changes in the gut microbiome of younger individuals that are indicative of metabolic perturbation [190]. Although evidence is lacking to indicate whether sleep restoration may help to improve gut microbiome health, it is worthy to note that sleep habits are demonstrated to influence myofibrillar protein synthesis [192], and that poor sleep duration and/or quality has been linked to an increased likelihood of sarcopenia [193].

### 3.4. Polypharmacy

More than 90% of older adults (≥65 years) regularly take at least one medication, and 57% of older women and 44% of older men report using 5 or more medications weekly (i.e., polypharmacy) [194]. Polypharmacy and the use of inappropriate medications in older individuals is independently associated with functional impairment [195] and a decline in activities of daily living [196] that may me mediated, at least in part, through drug-microbiome relationships. Medications and the microbiome have bidirectional interactions; medications may elicit inhibitory or beneficial effects on gut microbes, meanwhile, gut microbes play an active role in drug metabolism [197]. Though a complete understanding of the impact of polypharmacy on aging microbial reassembly is lacking, associations between medication use and bacterial dysbiosis, reduced biodiversity, and a reduction in SCFA producing taxa (e.g., *Prevotella* and *Parabacteroides*) in hospitalized older adults [198] portend to the deleterious impact of excessive medication use on microbial homeostasis and potentially anabolic responsiveness (Figure 2). 

The rapid and transformative effects of antibiotic exposure on microbiome composition, and the health implications of these alterations, have received the greatest proportion of scientific inquiry in this area. Broad-spectrum antibiotics can affect the abundance of ~30% of gut bacteria [199], and these antibiotic-induced microbiota alterations can persist for years after the cessation of antibiotic treatment [200]. Chief among the documented dysbiotic effects of antibiotic therapy is a reduction in microbial diversity and an attenuation of SCFA’s [201], both of which are hypothesized to play an integral role in anabolic potential. Recent *in vitro* work set out to examine the impact of non-antibiotic therapies on human microbiome composition, and nearly a quarter of more than 1,000 marketed drugs were demonstrated to inhibit the growth of at least one bacterial strain [202]. Taken together, these studies highlight the extensive impact of drug therapies on microbiome structure, and the importance of considering polypharmacy as a contributor to reductions in microbial homeostasis with advancing age.

## 4. Perspectives and Future Directions

A bidirectional gut-muscle axis with implications for aging skeletal muscle size, quality, and function has been proposed [16,17,18,19,20,21]. Extending on this aging gut-muscle axis, we propose that age-related changes in gut microbiota may detract from the anabolic response of skeletal muscle to protein feeding in older adults. Intriguingly, many of these adverse microbial modifications seem to be avoided in long-lived models of highly successful aging. Through the above, we describe how commonly observed age-related changes in the gut microbiome may compromise anabolic responsiveness through their impact on protein digestion and amino acid absorption, circulating amino acid availability, anabolic hormone production and responsiveness, and anabolic intramuscular signaling. While some of these age-associated gut microbiome alterations may simply be a product of the natural aging process, we believe that lifestyle modification (i.e., improved diet, exercise, sleep, and reducing medication use) may help to preserve gut microbial equilibrium in a manner that would be anticipated to maintain anabolic capacity in older years. To validate this hypothesis, interventional studies attempting to manipulate microbial ecology as a means to potentiate muscle protein synthetic responses to anabolic stimuli (i.e., protein feeding) in older individuals are needed. Given the pertinence of maintaining microbial homeostasis for the promotion of skeletal muscle and whole-body health, exploration of novel lifestyle therapies (e.g., probiotics, resistance training, sleep therapy) to promote healthy aging of the gut microbiome is also of interest.

## Figures and Tables

**Figure 1 nutrients-13-00706-f001:**
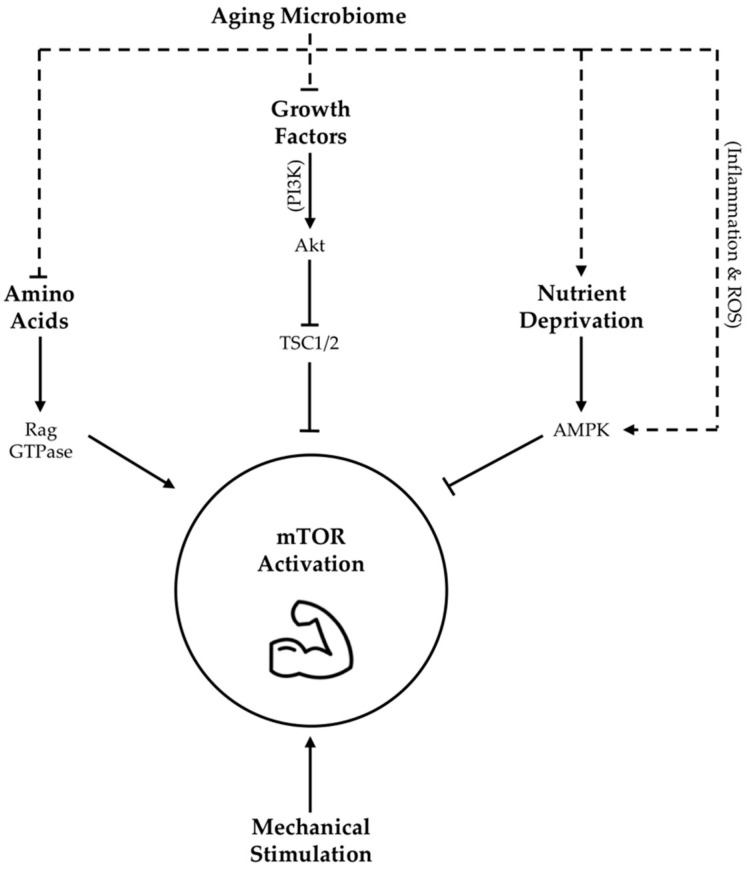
**Proposed influence of age-related changes in gut microbiota on skeletal muscle mTOR signaling**. Dotted lines indicate hypothesized contribution of aging microbiome changes to upstream regulators of mTOR activity, while solid lines indicate contributions as per literature consensus. TSC1/2 = tuberous sclerosis complex 1/2; Akt = protein kinase B, also abbreviated as PKB; AMPK = adenosine monophosphate activated protein kinase; ROS = reactive oxygen species; PI3K = phosphoinositide-3 kinase; GTPase = guanosine triphosphate-ase.

**Figure 2 nutrients-13-00706-f002:**
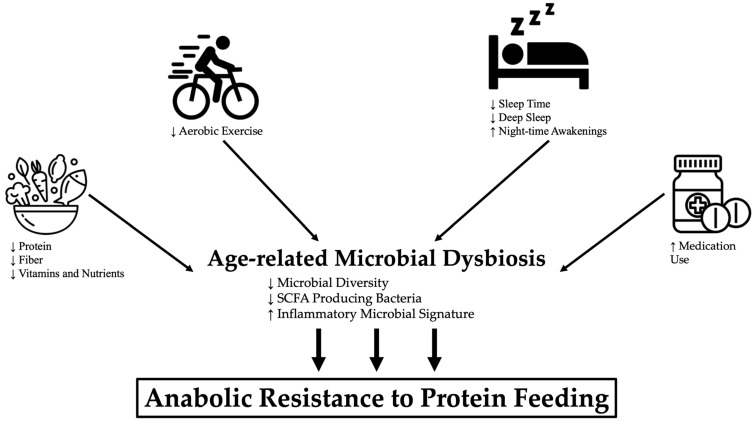
**Downstream effects of modifiable lifestyle factors on microbial dysbiosis and anabolic resistance.** Deleterious changes in modifiable lifestyle factors (e.g. diet, exercise, sleep, polypharmacy) commonly observed with aging promote microbial dysbiosis and consequential anabolic resistance to protein feeding. ↑ = increased; ↓ = decreased

**Table 1 nutrients-13-00706-t001:** The Aging Gut Microbiome & Anabolic Sensitivity.

Biological Process	Age-Related Change	Suspected Gut Microbiota Contribution
**Protein Digestion & ** **Amino Acid Absorption**	Slowed protein digestion/absorption	- Microbial-mediated alterations in activity of digestive enzymes such as aminopeptidase N [31,32]
	Maladaptive proteolytic fermentation	- ↑ protein fermenting bacteria = ↑ toxic microbial byproducts (hydrogen sulfide, ammonia, *p*-cresyl, and indoxyl sulfate) [23,33,34,35] - ↓ butyrogenic microflora = ↓ short chain fatty acids and biogenic amines [36,37]
**Circulating Amino ** **Acid Availability**	Reduced microbial-derived amino acid biosynthesis	- ↓ bacteria involved in lysine biosynthesis [38,39,40,41] - ↓ bacteria involved in leucine biosynthesis [42,43]
	Splanchnic extraction of dietary amino acids	- ↑ leucine oxidation due to gut dysbiosis [44,45,46,47]
**Anabolic Hormone ** **Production & Responsiveness**	Impaired insulin-mediated skeletal muscle perfusion	- ↑ pro-inflammatory microbes [48,49] - ↑ toxic microbial metabolites (e.g., trimethylamine-*N*-oxide (TMAO), indoxyl sulfate) = ↓ nitric oxide (NO) bioavailability and endothelial function [49,50,51,52,53]
	Decrease in growth hormone (GH)/insulin-like growth factor 1 (IGF-1) axis activity	- ↓ IGF-1 supportive bacterial species (e.g. *Lactobacillus plantarum*) [54,55,56,57,58,59,60,61]
**Intramuscular Anabolic** **Signaling Response**	Mammalian target of rapamycin (mTOR) inhibited by amino acid/growth factor suppression	- ↑ pro-inflammatory microbes = ↓ insulin sensitivity = ↓ IGF-1 activity [62] - ↓ nutrient absorption = ↓ mTOR signaling [63] - ↓ amino acid availability = blunted mTOR activity [64]
	mTOR inhibited by adenosine monophosphate activated protein kinase (AMPK) upregulation	- ↑ LPS = ↑ pro-inflammatory cytokines = ↑ AMPK activation [65,66,67,68] - Dysbiotic microbial signaling = ↑ reactive oxygen species (ROS) = ↑ chronic inflammation = ↑ AMPK activation [69,70,71]

↑= increased; ↓ = decreased.

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
