# Peer review of "Evidence for the Contribution of Gut Microbiota to Age-Related Anabolic Resistance"

_nutrients, 2021, doi:10.3390/nu13020706_

Round 1

Reviewer 1 Report

Dear Authors,

The written organized way need to be improved, it is difficult to follow and the reader often gets lost in reading complex mechanisms whose description, in many cases, is already known (e.g. Intramuscular Signaling) and does not add new details to scientific knowledge. There are also entire paragraphs in which it is not clear what is the possible link between the elements described and the topic of the work.

I think this manuscript might be interesting in the original idea, but it needs to be completely improved.

Best regards

Author Response

Dear Authors,

The written organized way need to be improved, it is difficult to follow and the reader often gets lost in reading complex mechanisms whose description, in many cases, is already known (e.g. Intramuscular Signaling) and does not add new details to scientific knowledge. There are also entire paragraphs in which it is not clear what is the possible link between the elements described and the topic of the work.

I think this manuscript might be interesting in the original idea, but it needs to be completely improved.

We thank the reviewer for their interest in this exciting topic and have made substantial revisions so as to greatly enhance the organizational structure of our review. As recommended by the reviewer, this involved elimination of some relevant but non-essential information regarding mechanisms underlying the biological processes governing skeletal muscle protein synthesis. For example, in section 2.4 (Anabolic Hormone Responses) an entire paragraph describing the structure and regulation of growth hormone and IGF-1 was removed (lines 284-288). Similarly, in section 2.5 (Intramuscular Signaling) an entire paragraph was removed (lines 361-371) and some background historical perspective was deleted (lines 239-240). We also eliminated interesting but not overly necessary suggestions for future research (lines 193-196).

In addition, per the recommendation of reviewer #3, we have added 1 table and 2 figures so as to increase the accessibility of our message.

We believe these major revisions have made our work much easier to follow and are appreciative of the reviewer’s criticism.

Reviewer 2 Report

It will also be interesting to discuss the gut microbiota in long-lived individuals. Aging is very heterogeneous and the influence of cultural and environmental factors on diet and epigenetics will need to be carefully considered.

Author Response

It will also be interesting to discuss the gut microbiota in long-lived individuals. Aging is very heterogeneous and the influence of cultural and environmental factors on diet and epigenetics will need to be carefully considered.

We thank the reviewer for their enthusiastic support of our work and appreciate this insightful recommendation. Indeed, recent studies in centenarians have demonstrated unique microbial characteristics that provide fascinating perspectives into the biological vs. environmental determinants of healthy aging. In the context of the present work, we found papers by Kong et al. (PMID: 27676296), Wang et al. (PMID: 26003628), and Biagi et al. (PMID: 28049008) to be particularly insightful, and have integrated information from these works, and others, throughout our manuscript. Specific edits include:

  • Lines 68-70: “Along the way, the gut microbiota of long-lived individuals (e.g., centenarians) will be considered as this population provides researchers with an insightful model of healthy aging
  • Lines 85-88: “and compare these age-related changes with those observed in healthy, long-lived individuals, an increasingly explored population that may provide novel perspective regarding the maintenance of anabolic sensitivity and muscle mass.”
  • Lines 158-160: “Intriguingly, several reports examining the microbiome of healthy, long-lived animal and human models (≥90 yrs) have observed microbial signatures containing elevated levels of SCFA producers [54-56].
  • Lines 175-176: “yet are observed to increase in healthy 90+ year old individuals [55,56,62-65]
  • Lines 274-277: “In comparison, recent work by Hallberg et al. [116] noted that healthy centenarians exhibit a greater abundance of Eubacterium limosum, which have the potential to demethylate TMA, preventing its conversion to circulating TMAO.
  • Lines 329-332: “Moreover, a recent systematic review underscored the greater abundance of plantarum in healthy long-lived individuals [141], speculatively highlighting the importance of L. plantarum preservation throughout aging.
  • Lines 399-402: “However, it is interesting to note that LPS stimulation seems to evoke less of an inflammatory response in long-lived compared to normally age mice [158], alluding to the importance of heightened antioxidant defense mechanisms as a characteristic of highly successful aging.
  • Line 427: “which tend to be enriched in healthy long-lived individuals [64,65,141]
  • Lines 518-520: “Intriguingly, many of these adverse microbial modifications seem to be avoided in long-lived models of highly successful aging

Reviewer 3 Report

Overall, this review article is well written and structured, but a couple of my concerns are:

  1. In a review article, a pictorial or graphical representation is more appealing than the text part. In order to improve the quality of the article, the authors should include a few more figures and/table.
  2. It will have an additional impact on the manuscript if a brief mechanism of skeletal muscle loss and/or sarcopenia is included as a sub-heading.
  3. line number 12-13, "Building on this work, this review will examine" it is not much clear to me.
  4. The abstract is short, it should be updated specifically by outlining the need for this review, the scope, and the significance of this review.

Author Response

Overall, this review article is well written and structured, but a couple of my concerns are:

We thank the reviewer for their complimentary feedback regarding the writing style and structure of our work, and have responded to particular concerns below in a point-by-point manner.

  1. In a review article, a pictorial or graphical representation is more appealing than the text part. In order to improve the quality of the article, the authors should include a few more figures and/table.

We thank the reviewer for this recommendation and agree that tables and figures are particularly useful in a review article. To this point, we have added a table (page 3) summarizing the key points and associated citations in the body of the text (Table 1. The Aging Gut Microbiome & Anabolic Sensitivity), and have replaced some non-essential text with two additional figures. Figure 1 (page 10) depicts the proposed influence of age-related changes in gut microbiota on skeletal muscle mTOR signaling. Figure 2 (page 13) illustrates the downstream effects of modifiable lifestyle factors on microbial dysbiosis and anabolic resistance. The initial figure from our first submission has been removed and now will be used as a graphical abstract.

  1. It will have an additional impact on the manuscript if a brief mechanism of skeletal muscle loss and/or sarcopenia is included as a sub-heading.

Owing to the demonstrable impact of sarcopenia in a rapidly expanding aging demographic, we agree that greater emphasis on this point is warranted. To this point, we have revised our manuscript in the following manners:

  • Inserted new sub-heading (line 76) titled “2.1 Overview of the Aging Microbiome, Anabolic Resistance, & Sarcopenia
  • lines 37-38: “a more definitive understanding of the biological mechanisms underlying sarcopenia is warranted.”
  • lines 61-63: “The purpose of this review will be to provide evidence in support of the hypothesis that age-associated changes in gut microbiota composition contribute to anabolic resistance following protein feeding in older adults that underlie sarcopenia.

  1. line number 12-13, "Building on this work, this review will examine" it is not much clear to me.

We apologize for the confusion and lack of clarity and have revised this sentence as such (lines 20-23): “This review will examine how age-related changes in gut microbiota composition may impact anabolic response to protein feeding through adverse changes in protein digestion and amino acid absorption, circulating amino acid availability, anabolic hormone production and responsiveness, and intramuscular anabolic signaling.

  1. The abstract is short, it should be updated specifically by outlining the need for this review, the scope, and the significance of this review.

We agree that the abstract could be expanded upon to highlight the critical need for research examining mechanisms of aging muscle loss and sarcopenia. In consideration of this recommendation, we have updated our abstract as follows: “Globally, people 65 years of age and older are the fastest growing segment of the population. Physiological manifestations of the aging process include undesirable changes in body composition, declines in cardiorespiratory fitness, and reductions in skeletal muscle size and function (i.e., sarcopenia) that are independently associated with mortality. Decrements in muscle protein synthetic responses to anabolic stimuli (i.e., anabolic resistance), such as protein feeding or physical activity, are highly characteristic of the aging skeletal muscle phenotype and play a fundamental role in the development of sarcopenia. A more definitive understanding of the mechanisms underlying this age-associated reduction in anabolic responsiveness will help to guide promyogenic and function promoting therapies. play a fundamental role in the age-associated decline of skeletal muscle size and function (sarcopenia). Recent studies have provided evidence in support of a bidirectional gut-muscle axis with implications for aging muscle health and sarcopenia. Building on this work, tThis review will examine how age-related gut dysbiosis changes in gut microbiota composition may impact anabolic response to protein feeding through adverse changes in protein digestion and amino acid absorption, circulating amino acid availability, anabolic hormone production and responsiveness, and intramuscular anabolic signaling. We conclude by reviewing literature describing lifestyle habits suspected to contribute to age-related changes in the microbiome with the goal of identifying evidence-informed strategies to preserve microbial homeostasis, anabolic sensitivity, and skeletal muscle with advancing age. that may be modified by older adults in an attempt to maintain gut microbial homeostasis with the hopes of preserving anabolic responsiveness and skeletal muscle health throughout the lifespan.

Round 2

Reviewer 1 Report

Dear authors,

thank you for the substantial revision of the manuscript.

I think the work have be improved.

Best regards